

# Global symmetry and conformal bootstrap
# in the two-dimensional Q-state Potts model

Rongvoram Nivesvivat⋆

Institut de physique théorique, CEA, CNRS, Université Paris-Saclay,
Gif-sur-Yvette 91191, France

⋆ rongvoram.nivesvivat@ipht.fr

## Abstract

The Potts conformal field theory is an analytic continuation in the central charge of conformal field theory describing the critical two-dimensional $Q$-state Potts model. Four-point functions of the Potts conformal field theory are dictated by two constraints: the crossing-symmetry equation and $S_Q$ symmetry. We numerically solve the crossing-symmetry equation for several four-point functions of the Potts conformal field theory for $Q \in \mathbb{C}$. In all examples, we find crossing-symmetry solutions that are consistent with $S_Q$ symmetry of the Potts conformal field theory. In particular, we have determined their numbers of crossing-symmetry solutions, their exact spectra, and a few corresponding fusion rules. In contrast to our results for the $O(n)$ model, in most of examples, there are extra crossing-symmetry solutions whose interpretations are still unknown.

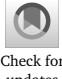

# 1 Introduction

The two-dimensional $Q$-state Potts model is well-known to become conformally invariant at the critical temperature for $0 \le Q \le 4$. At the critical point, the Potts model can be described by conformal field theory (CFT) whose central charge $c$ is related to $Q$ by the $\beta^2$-parametrization:

$$Q = 4\cos^2(\pi\beta^2), \quad \text{with} \quad \frac{1}{2} \le \beta^2 \le 1, \quad c = 13 - 6\beta^2 - 6\beta^{-2}. \tag{1}$$

While the Potts model with $Q \in \mathbb{N} + 2$ is notable for generalizing the Ising model in two dimensions, the case of generic $Q$ represents the so-called Fortuin-Kasteleyn random clusters in which $Q$ only appears as a formal parameter in correlation functions and is no longer required to be an integer [1]. With the latter description, correlation functions of the lattice model exist for $Q \in \mathbb{C}$ [2]. Using (1), it therefore makes sense to expect CFT describing the critical $Q$-state Potts model to be consistent at generic central charge as well. This motivates us to define the Potts conformal field theory as follows [3]:

*The Potts conformal field theory is an analytic continuation in the central charge $c$ of the critical $Q$-state Potts model [4] such that*

$$\Re(c) < 13 \iff \Re(\beta^2) > 0. \tag{2}$$

More precisely, the Potts conformal field theory is a family of CFTs, which is parametrized by the parameter $\beta^2$, that is to say the Potts CFT lives on the $\beta^2$-half plane, or equivalently on the double cover of $c$-half plane. For some special values of $\beta^2$, the Potts CFT describes some well-known models in statistical physics. For instance,

| $Q$ | $c$ | $\beta^2$ | The Potts CFT |
|-----|-----|-----------|---------------|
| 0 | $-2$ | $\frac{1}{2}$ | the critical spanning trees |
| 1 | 0 | $\frac{2}{3}$ | the critical bond percolation |
| 2 | $\frac{1}{2}$ | $\frac{3}{4}$ | the critical Ising model |

$$\tag{3}$$

In the case of the critical Ising model, we stress here that the Potts CFT describes observables of Fortuin-Kasteleyn random clusters and is therefore not the Ising minimal model in [5]. However, we do not know yet the statistical interpretation of the Potts CFT for generic $\beta^2 \in \mathbb{C}$. Thus, the Potts CFT should be considered as a theory that includes the Potts model as special cases.

The critical $Q$-state Potts model does not only have local conformal symmetry but also $S_Q$ symmetry as global symmetry, whose representation theory can be formulated as tensor categories for non-integer $Q$ [6]. As CFT data, the Potts CFT is therefore a collection of primary

fields and their operator-product expansion (OPE) coefficients which satisfy the consistency conditions: the crossing-symmetry equation and constraints from $S_Q$ symmetry. In [4], the authors obtained a list of primary fields in the Potts CFT by computing the torus partition function. The complete action of the Virasoro algebra and $S_Q$ symmetry on these primary fields was recently determined in [7] and [8], respectively. The next step in solving the Potts CFT is then to solve the crossing-symmetry equation for their OPE coefficients. Numerically, this can be done by using the conformal bootstrap approach, initially proposed in [9] wherein the crossing-symmetry equation is considered as linear equations for four-point structure constants.

In recent years, much of interest has been focusing on solving the simplest non-trivial four-point functions of the Potts CFT, namely the four-point connectivities. The four-point connectivities compute the probability of how the four points belong to the Fortuin-Kasteleyn clusters. There are four different configurations of these connectivities [10], namely $P_{aaaa}$, $P_{abab}$, $P_{aabb}$, and $P_{abba}$, which can be represented as follows:

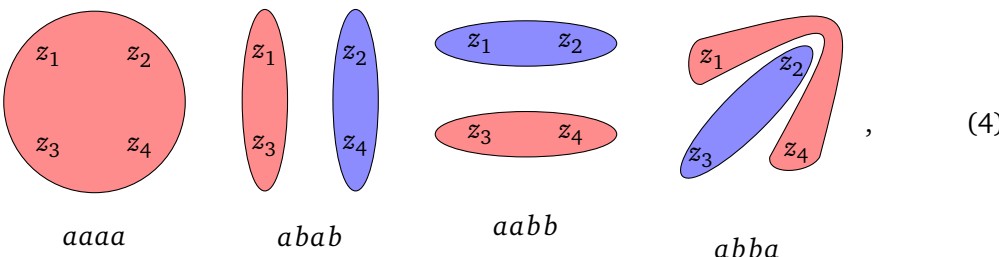

$$ , \qquad (4)$$

where different colors indicate different Fortuin-Kasteleyn clusters. Their spectra were completely determined in [2] by using the transfer-matrix method on the lattice model and have also been validated by the numerical conformal bootstrap in [11] and [7]. Furthermore, the authors of [11] also found several analytic ratios of some OPE coefficients in these connectivities, which suggest the possibility of exact solutions to the Potts CFT.

Other four-point functions however have been left relatively uncharted. In this article, we start the exploration of four-point functions of arbitrary primary fields in the Potts CFT by using the method of [12], originally introduced for the $O(n)$ CFT. The outline is as follows: we review the spectrum of the Potts CFT in Section 2, then we explain how to solve the crossing-symmetry equation numerically and define four-point functions of the Potts CFT in Section 3, and we demonstrate how to compute numerically four-point functions of the Potts CFT for several examples in Sections 4 and 5.

**Main results**

Let us also highlight results that we consider interesting.

- In Section 3.2, we define four-point functions of the Potts CFT as crossing-symmetry solutions that transform in the $S_Q$ representations. In particular, these solutions must obey the constraint (34).

- In Section 4, we solve the crossing-symmetry equation with the full spectrum of the Potts CFT [4] for several four-point functions, however we find that there are extra solutions, which do not satisfy the constraint (34) and therefore cannot belong to the Potts CFT. Detailed discussion of how to single out these extra solutions can be found in Section 5.

- On tables in Section 4.3, we display the numbers of solutions, obtained by solving the crossing-symmetry equation for 28 four-point functions with the full spectrum of the Potts CFT [4]. These tables include both the numbers of total solutions and the numbers of solutions which obey (34).

- At the end of Section 4.3, we conjecture a relation between the numbers of crossing-symmetry solutions and the existence of the degenerate fields in the spectra of four-point functions for both the Potts CFT and the $O(n)$ CFT [12].

- Based on several examples of Section 4, we propose the fusion rules of $V_{(0,\frac{1}{2})} \times V_{(0,\frac{1}{2})}$, $V_{(0,\frac{1}{2})} \times V_{(2,\frac{1}{2})}$, and $V_{(2,0)} \times V_{(0,\frac{1}{2})}$. These fusion rules also led us to a conjecture for vanishing three-point functions (53).

Numerical data for this article can be found in the notebook `Potts4pt.ipynb` in [13].

## 2 Spectrum of the model

Conformal dimensions of primary fields in the Potts CFT are characterized by the Kac indices,

$$\Delta_{(r,s)} = P_{(r,s)}^2 - P_{(1,1)}^2, \quad \text{with} \quad P_{(r,s)} = \frac{1}{2}\left(r\beta - \frac{s}{\beta}\right), \tag{5}$$

where the indices $r$ and $s$ always take rational values, and the parameter $\beta$ is defined in (2). From [4], the list of primary fields in the Potts CFT reads

$$\mathcal{Z}^{\text{Potts}} = \left\{V_{\langle 1,s\rangle}^D\right\}_{s\in\mathbb{N}^*} \cup \left\{V_{(0,s)}\right\}_{s\in\mathbb{Z}+\frac{1}{2}} \cup \left\{V_{(r,s)}\right\}_{\substack{r\in\mathbb{N}+2 \\ s\in\frac{\mathbb{Z}}{r}}}. \tag{6}$$

The field $V_{\langle 1,s\rangle}^D$ is a degenerate-diagonal primary field with the conformal dimensions $(\Delta, \bar\Delta) = (\Delta_{(1,s)}, \Delta_{(1,s)})$, and we write $V_{(r,s)}$ for a non-diagonal primary field whose left and right dimensions are given by $(\Delta, \bar\Delta) = (\Delta_{(r,s)}, \Delta_{(r,-s)})$, including the case $r = 0$. Furthermore, the degenerate fields $V_{\langle 1,s\rangle}^D$ in the spectrum (6) come with multiplicity one, while the non-diagonal primary fields $V_{(0,s)}$ and $V_{(r,s)}$ have the multiplicities:

$$L_{(0,s)} = Q - 1, \tag{7}$$

$$L_{(r,s)}(Q) = (Q-1)(-1)^r \delta_{s\in\mathbb{Z}+\frac{r+1}{2}} + \frac{1}{r}\sum_{r'=0}^{r-1} e^{2\pi i r's} p_{r\wedge r'}(Q-2), \tag{8}$$

where $r \wedge r'$ denotes the greatest common divisor of $r$ and $r'$, and the functions $p_d(x)$ are the modified Chebyshev polynomials, defined by the recursion:

$$x p_d(x) = p_{d-1}(x) + p_{d+1}(x), \quad \text{with} \quad p_1(x) = x \quad \text{and} \quad p_0(x) = 2. \tag{9}$$

For instance, we have

$$p_1(x) = x, \tag{10a}$$
$$p_2(x) = x^2 - 2, \tag{10b}$$
$$p_3(x) = x(x^2 - 3), \tag{10c}$$
$$p_4(x) = x^4 - 4x^2 + 2. \tag{10d}$$

From the formula (8), $L_{(r,s)}$ are invariant under the shifts,

$$s \to s + \mathbb{Z} \quad \text{and} \quad s \to -s. \tag{11}$$

It is therefore sufficient to compute $L_{(r,s)}$ for $0 \leq s < 1$. For example,

$$L_{(2,0)} = \frac{Q}{2}(Q-3),\tag{12a}$$

$$L_{(2,\frac{1}{2})} = \frac{1}{2}(Q-1)(Q-2),\tag{12b}$$

$$L_{(3,0)} = \frac{1}{3}(Q-1)(Q^2-5Q+3),\tag{12c}$$

$$L_{(3,\frac{1}{3})} = \frac{Q}{4}(Q-2)(Q-4),\tag{12d}$$

$$L_{(4,0)} = \frac{Q}{4}(Q-2)(Q-3)^2.\tag{12e}$$

Notice that all coefficients of polynomials in the examples (12) are rational numbers, which is not obvious from the expression (8) because of the factor $e^{2\pi i r' s}$. It was, however, recently proven in [8] that the multiplicities $L_{(r,s)}$ are always polynomials in $Q$ with rational coefficients. Moreover, these non-trivial multiplicities reflect the fact that primary fields in the Potts CFT also transform in irreducible representations of $S_Q$ symmetry. For example, it was first observed in [14] that the multiplicities $L_{(r,s)}$ can always be written as a sum of the dimensions of $S_Q$ irreducible representations with positive integer coefficients.

## 2.1 Action of the Virasoro algebra

At generic central charge, the Virasoro algebra acts on primary fields in the full spectrum (6) as follows:

- The diagonal primary fields $V_{\langle 1,s \rangle}^D$ with $s \in \mathbb{N}^*$ belong to the degenerate representations of the Virasoro algebra and come with one vanishing null descendant at level $s$ [11,15]. For example, the identity field $V_{\langle 1,1 \rangle}^D$ has $L_{-1}V_{\langle 1,1 \rangle}^D = 0$ as its vanishing null descendant.

- The non-diagonal primary fields $V_{(r,s)}$ with both $r, s \in \mathbb{Z} - \{0\}$ transform in the logarithmic representations $\mathcal{W}_{(r,s)}^\kappa$ of [7]. These representations lead to a second-rank Jordan cell of the Virasoro-generator $L_0$ and are parametrized by the logarithmic coupling $\kappa$, which was determined for any $r$ and $s$ in [7].

- The other non-diagonal primary fields $V_{(r,s)}$ with $s = 0$ or $s \notin \mathbb{Z}$ belong to Verma modules.

It is also worth mentioning that four-point conformal blocks of these representations have been determined analytically. The expressions for the conformal blocks of degenerate representations and Verma modules are well known, given by the so-called Zamolodchikov recursion [16]. For the logarithmic case, conformal blocks with primary fields as external fields have been completely determined in [7].

## 2.2 Action of $S_Q$ symmetry

Irreducible representations of symmetric group $S_Q$ can be parametrized by Young diagram whose number of boxes is $Q$ [17]. We denote Young diagrams by decreasing sequences of positive integers in which each integer indicates number of boxes in each row. For example, the sequence $[7, 5, 3, 2, 2]$ represents the following diagram,

$$[\lambda_0] = [7, 5, 3, 2, 2] = [75322] = \quad, \quad \text{with} \quad |\lambda_0| = 7+5+3+2+2 = 19,\tag{13}$$

where $|\lambda_0|$ is the size of the diagram $[\lambda_0]$. Moreover, we always neglect writing commas in Young diagrams, whenever there is no ambiguity. Irreducible representations of $S_Q$ with $Q \in \mathbb{C}$ can then be labelled as the Young diagrams $[\lambda]$, obtained by removing the first row of the Young diagrams $[Q - |\lambda|, \lambda]$ that parametrize irreducible representations of symmetric group $S_Q$ [18, 19]. Thus, the resulting Young diagrams are independent of $Q$. For instance,

| $S_Q$ reps | Integer $Q$ | $Q \in \mathbb{C}$ |
|---|---|---|
| singlet | $[Q]$ | $[\,]$ |
| fundamental | $[Q-1,1]$ | $[1]$ |
| symmetric | $[Q-2,2]$ | $[2]$ |
| anti-symmetric | $[Q-2,1,1]$ | $[11]$ |

$$(14)$$

From [4], the degenerate fields $V^D_{\langle 1,s \rangle}$ transform under $S_Q$ as the singlet, while the non-diagonal fields $V_{(0,s)}$ belong to the fundamental representation. We denote them as follows,

$$\Lambda_{\langle 1,s \rangle^D} = [\,] \quad \text{and} \quad \Lambda_{(0,s)} = [1]. \tag{15}$$

From the twisted-torus partition function in [8], the other non-diagonal fields $V_{(r,s)}$ transform under $S_Q$ as the representations $\Lambda_{(r,s)}$ whose expressions are given by

$$\Lambda_{(r,s)} = (-1)^r \delta_{s \in \mathbb{Z} + \frac{r+1}{2}}[1] + \frac{1}{r} \sum_{r'=0}^{r-1} e^{2\pi i r's} p_{r \wedge r'} \left( \sum_{r'' | \frac{r}{r \wedge r'}} \Lambda_{r''} - 2[\,] \right), \tag{16}$$

where $\Lambda_r$ are formal representations of $S_Q$ defined by

$$\Lambda_r = [\,] + \sum_{k=0}^{r-1} (-1)^k [r-k, 1^k], \quad \text{with} \quad \dim(\Lambda_1) = Q \quad \text{and} \quad \dim(\Lambda_{r \geq 2}) = 0. \tag{17}$$

From (8) and (16), the dimension of $\Lambda_{(r,s)}$ always matches $L_{(r,s)}$. The authors of [8] have also checked extensively, in a number of examples, that the formula (16) always yields a sum of $S_Q$ irreducible representations with positive integer coefficients, ensuring us that $\Lambda_{(r,s)}$ are indeed $S_Q$ representations. To compute (16), recall the tensor products of $S_Q$ irreducible representations for $Q \in \mathbb{C}$ [18]:

$$[\lambda] \times [\mu] = \sum_{\nu} M_{\lambda, \mu, \nu} [\nu], \tag{18}$$

where $M_{\lambda, \mu, \nu}$ are the reduced Kronecker coefficients, which do not depend on $Q$ and are strictly positive integers [20]. Furthermore, the sum of $S_Q$ representations in (18) is subject to the following constraint [21]:

$$M_{\lambda, \mu, \nu} \neq 0 \implies ||\lambda| - |\mu|| \leq |\nu| \leq |\lambda| + |\mu|. \tag{19}$$

There are simple rules of computing a product $[Q-1,1] \times [\mu]$ for symmetric group $S_Q$ in [17], which can be rewritten for the case $[\lambda] = [1]$ in (18) as follows: the product $[1] \times [\mu]$ is a sum of all possible Young diagrams obtained by removing one box from $[\mu]$, then adding at most one box to the resulting diagram where the multiplicity for each diagram is one except for the diagram $[\mu]$ itself whose coefficient is the number of different rows. For instance,

$$[21] \times [1] = 2[21] + [31] + [22] + [211] + [3] + [111] + [2] + [11]. \tag{20}$$

Using these rules, one can also obtain more general results by applying associativity. In practice, we have used a program written in SageMath by [22] to compute the product (18). Let us show a few more tensor products of $S_Q$ with $Q \in \mathbb{C}$:

$$[1] \times [1] = [1] + [2] + [11] + [], \tag{21a}$$

$$[2] \times [1] = [1] + [2] + [11] + [21] + [3], \tag{21b}$$

$$[11] \times [1] = [1] + [2] + [11] + [21] + [111], \tag{21c}$$

$$[2] \times [2] = [4] + [31] + [22] + [3] + 2[21] + [111] + 2[2] + [11] + [1] + []. \tag{21d}$$

Likewise to $L_{(r,s)}$, the representations $\Lambda_{(r,s)}$ are also invariant under (11). Thus, we only need to compute $\Lambda_{(r,s)}$ for $0 \le s < 1$. Let us now display some examples of $\Lambda_{(r,s)}$:

$$\Lambda_{(2,0)} = [2], \tag{22a}$$

$$\Lambda_{(2,\frac{1}{2})} = [11], \tag{22b}$$

$$\Lambda_{(3,0)} = [3] + [111], \tag{22c}$$

$$\Lambda_{(3,\frac{1}{3})} = [21], \tag{22d}$$

$$\Lambda_{(4,0)} = [4] + [22] + [211] + [3] + [21] + 2[2] + [1] + [], \tag{22e}$$

$$\Lambda_{(4,\frac{1}{4})} = [31] + [211] + [21] + [111] + [11], \tag{22f}$$

$$\Lambda_{(4,\frac{1}{2})} = [31] + [22] + [1111] + [3] + [21] + [2] + [11] + [1], \tag{22g}$$

$$\Lambda_{(5,0)} = [5] + [32] + 2[311] + [221] + [11111] + [4] + 3[31]$$
$$+ 2[22] + 3[211] + [1111] + 2[3] + 4[21] + 2[111] + 2[2] + 2[11] + [1]. \tag{22h}$$

The representations $\Lambda_{(r,s)}$ in (16) tell us how the non-diagonal fields $V_{(r,s)}$ transform under $S_Q$. For example, $\Lambda_{(3,0)}$ leads to two independent fields: $V_{(3,0)}^{[3]}$ and $V_{(3,0)}^{[111]}$. Let us then introduce further notations:

- $V_{(r,s)}^{\lambda}$ is a non-diagonal primary field that also transforms in the irreducible representation $\lambda$ of $S_Q$.

- $V_{(r,s)}^{\lambda}$ with multiplicity $a$ can be denoted by $V_{(r,s)}^{\lambda,i}$ for $i = 1, \dots, a$. For instance, from (22e), we have $V_{(4,0)}^{[2],1}$ and $V_{(4,0)}^{[2],2}$.

- $V^{\lambda}$ is a field that belongs to the irreducible representation $\lambda$ of $S_Q$.

## 3 Solving the crossing-symmetry equation

We explain how to numerically solve the crossing-symmetry equation for four-point functions of non-diagonal primary fields in the Potts CFT by using the approach of [12], whereas four-point functions whose external fields involve at least one degenerate field are known to satisfy the BPZ equations and can be determined analytically [23, 24].

### 3.1 Set-up

We shall decompose four-point functions of the Potts CFT into the so-called interchiral blocks, rather than the usual conformal blocks [11]. Interchiral blocks are universal objects which can be completely determined by conformal symmetry and the degenerate fields. For example, in the Potts CFT, the existence of the degenerate fields $V_{(1,s)}^D$ in (6) imply analytic ratios

between four-point structure constants of two primary fields (diagonal or not) with the indices $(r,s)$ and $(r,s+2\mathbb{Z})$, within the same four-point function [24]. Such relations then glue their corresponding conformal blocks together into an interchiral block. Schematically,

$$
\text{(diagram)} = \sum_{j\in 2\mathbb{Z}} \frac{D_{(r,s+j)}}{D_{(r,s)}} \text{(diagram)} , \tag{23}
$$

$$
\text{Interchiral block of } (r,s) \qquad\qquad \text{Conformal blocks of } (r,s+j)
$$

where interchiral blocks of any primary fields in (6) have been completely determined in [12]. Let us also briefly explain how to compute the ratios of structure constants in (23). These ratios can be obtained as products of three-point structure constants, which can be completely determined by using the BPZ equation, the crossing-symmetry equation, and the single-valuedess of four-point functions of the types: $\langle V^D_{\langle 1,2\rangle}V_1V_2V_3\rangle$ and $\langle V^D_{\langle 1,2\rangle}V_1V^D_{\langle 1,2\rangle}V_1\rangle$. For instance, see [24] for more details.

In addition, in the case of the four-point function $\langle V_{(0,\frac{1}{2})}V_{(0,\frac{1}{2})}V_{(0,\frac{1}{2})}V_{(0,\frac{1}{2})}\rangle$, we can further write the interchiral blocks of $(r,s)$ and $(r,s+1)$ as one interchiral block since ratios of structure constants in (23) factorize into a product of analytic ratios of structure constants which differ by one in the second indices [11]. However, without introducing any inconsistency, we shall always write interchiral blocks as in (23), to keep our set-up compatible with more general four-point functions.

Let us now write down the crossing-symmetry equation for four-point functions of primary fields in (6):

$$
\sum_{V\in\mathcal{S}^{(s)}} D^{(s)}_V \text{(diagram)} = \sum_{V\in\mathcal{S}^{(t)}} D^{(t)}_V \text{(diagram)} = \sum_{V\in\mathcal{S}^{(u)}} D^{(u)}_V \text{(diagram)} , \tag{24}
$$

$$
s\text{-channel} \qquad\qquad t\text{-channel} \qquad\qquad u\text{-channel}
$$

where $D^{(s)}_V$, $D^{(t)}_V$ and $D^{(u)}_V$ are the unknown four-point structure constants. We also stress here that it is necessary to solve the equation (24) simultaneously in all three channels to avoid having infinitely many solutions, whose interpretation is still an open problem [12].

The spectra $\mathcal{S}^{(s)}$, $\mathcal{S}^{(t)}$, and $\mathcal{S}^{(u)}$ in (24) are the full spectrum of the Potts CFT, allowed by conformal symmetry and the degenerate fields. Therefore, this set up gives us at least all crossing-symmetry solutions for each four-point function of the Potts CFT, as will be demonstrated for several examples in Sections 4 and 5. More precisely, since we are using the interchiral blocks, the spectrum of each channel in (24) is therefore the list of all primary fields in (6) modulo the degenerate fusion rules $V^D_{\langle 1,s\rangle}$ in [25]:

$$
V_{(r_0,s_0)} \times V^D_{\langle 1,s\rangle} = \sum_{\substack{j=s_0-s+1}}^{s_0+s-1} V_{(r_0,j)} . \tag{25}
$$

For example, the spectra for all three channels of $\langle V_{(0,\frac{1}{2})}V_{(0,\frac{1}{2})}V_{(0,\frac{1}{2})}V_{(0,\frac{1}{2})}\rangle$ are

$$
\mathcal{S}^{\text{Potts}} = \{(r,s)\in(\mathbb{N}+2)\times(-1,1]|rs\in\mathbb{Z}\}\cup\{(0,1/2)\}\cup\{\langle 1,1\rangle^D,\langle 1,2\rangle^D\}, \tag{26}
$$

where we always denote spectra of four-point functions by indices of their primary fields: $(r,s)$ for the non-diagonal fields $V_{(r,s)}$, and $\langle r,s \rangle^D$ for the degenerate fields $V_{\langle r,s \rangle}^D$. The fusion rules (25) allow both of the degenerate fields $V_{\langle 1,1 \rangle}^D$ and $V_{\langle 1,2 \rangle}^D$ to appear in (26) because of the coincidence,

$$V_{(0,\frac{1}{2})} = V_{(0,-\frac{1}{2})}. \tag{27}$$

That is to say we can write

$$V_{\langle 1,1 \rangle}^D \in V_{(0,\frac{1}{2})} \times V_{(0,\frac{1}{2})}, \tag{28a}$$

$$V_{\langle 1,2 \rangle}^D \in V_{(0,\frac{1}{2})} \times V_{(0,-\frac{1}{2})} = V_{(0,\frac{1}{2})} \times V_{(0,\frac{1}{2})}. \tag{28b}$$

Moreover, with the relation (27), any field of the type $V_{(0,s)}$ in (6) is also related to the field $V_{(0,\frac{1}{2})}$ by the shift $s \to s + 2\mathbb{Z}$. For instance, we have

$$V_{(0,\frac{3}{2})} = V_{(0,-\frac{1}{2}+2)}, \quad V_{(0,\frac{5}{2})} = V_{(0,\frac{1}{2}+2)}, \quad \text{and} \quad V_{(0,\frac{7}{2})} = V_{(0,-\frac{1}{2}+4)}. \tag{29}$$

Thus, the spectrum $\mathcal{S}^{\text{Potts}}$ is, in fact, the full spectrum (6) modulo the shift by two in the second indices. We also stress that, unlike $\langle V_{(0,\frac{1}{2})} V_{(0,\frac{1}{2})} V_{(0,\frac{1}{2})} V_{(0,\frac{1}{2})} \rangle$, the spectra $\mathcal{S}^{(s)}$, $\mathcal{S}^{(t)}$, and $\mathcal{S}^{(u)}$ for generic four-point functions are not always identical because of the degenerate fusion rules (25). For instance, the spectrum $\mathcal{S}^{(s)}$ of $\langle V_{(2,0)} V_{(2,0)} V_{(0,\frac{1}{2})} V_{(0,\frac{1}{2})} \rangle$ is $\mathcal{S}^{\text{Potts}} - \{\langle 1,2 \rangle^D\}$ whereas its $\mathcal{S}^{(t)}$ and $\mathcal{S}^{(u)}$ are $\mathcal{S}^{\text{Potts}} - \{\langle 1,1 \rangle^D, \langle 1,2 \rangle^D\}$.

**Numerical bootstrap**

Since the interchiral blocks in (24) are completely determined for any primary field in (6), the crossing-symmetry equation (24) is then a linear system for infinitely many unknown four-point structure constants, which can be numerically solved by using the method of [9]. In each spectrum of (24), the tower of infinitely many fields is truncated by an upper bound on their conformal dimensions,

$$\Re(\Delta + \bar{\Delta}) \leq \Delta_{\max}. \tag{30}$$

Computing the truncated crossing equation at random positions then gives us a linear system, from which we can solve for the four-point structure constants. The numerical error for each four-point structure constant, which we call the **deviation**, is given by the relative difference among structure constants of the same field, computed from different choices of positions. Recall that structure constants do not depend on positions, if the crossing-symmetry solutions converge, we then have

$$\text{deviation} \to 0, \quad \text{as} \quad \Delta_{\max} \to \infty. \tag{31}$$

See [9] and [7] for more details.

## 3.2 Four-point functions of the Potts CFT

The crossing-symmetry equation (24) only knows about conformal symmetry: representations of the Virasoro algebra and their conformal blocks. Four-point functions of the Potts CFT however also transform in irreducible representations of $S_Q$. These are two independent constraints which, in general, may not follow one another. Let us then discuss briefly how four-point functions of the Potts CFT are subject to $S_Q$ symmetry.

We begin with how $S_Q$ symmetry constrains two- and three-point functions of the Potts CFT. The Schur orthogonality relations infer

$$\nu \neq \mu \implies \langle V^\mu V^\nu \rangle = 0. \tag{32}$$

For three-point functions, the tensor product (18) implies

$$\nu \notin \lambda \times \mu \Longrightarrow \langle V^{\lambda} V^{\mu} V^{\nu} \rangle = 0 . \tag{33}$$

Notice that reversing the statements (32) and (33) does not always leads to correct results since two- and three-point functions are also constrained by conformal symmetry and OPE associativity. For instance, two-point functions of primary fields, which transform in the same $S_Q$ representations but have different conformal dimensions, vanish. We will also see similar situations for three-point functions in (53). Using the OPE, vanishing three-point functions in (33) then put constraints on the spectra of four-point functions of the Potts CFT, which led us to define four-point functions of the Potts CFT as follows:

*The four-point functions* $\langle \prod_{i=1}^{4} V_{(r_i,s_i)} \rangle$ *of the Potts CFT are solutions to the crossing-symmetry equation* (24) *whose spectra satisfy the constraints:*

$$\mathcal{S}^{(s)} \subset \mathcal{S}^{\Lambda_{(r_1,s_1)} \times \Lambda_{(r_2,s_2)}} \cap \mathcal{S}^{\Lambda_{(r_3,s_3)} \times \Lambda_{(r_4,s_4)}} , \tag{34a}$$

$$\mathcal{S}^{(t)} \subset \mathcal{S}^{\Lambda_{(r_1,s_1)} \times \Lambda_{(r_4,s_4)}} \cap \mathcal{S}^{\Lambda_{(r_2,s_2)} \times \Lambda_{(r_3,s_3)}} , \tag{34b}$$

$$\mathcal{S}^{(u)} \subset \mathcal{S}^{\Lambda_{(r_1,s_1)} \times \Lambda_{(r_3,s_3)}} \cap \mathcal{S}^{\Lambda_{(r_2,s_2)} \times \Lambda_{(r_4,s_4)}} , \tag{34c}$$

*where we have defined*

$$\mathcal{S}^{\sum_i \lambda_i} = \bigcup_i \mathcal{S}^{\lambda_i} , \quad with \quad \mathcal{S}^{\lambda} = \{ \kappa \in \mathcal{S}^{Potts} | \lambda \in \Lambda_{\kappa} \} . \tag{35}$$

The spectrum $\mathcal{S}^{\lambda}$ is a set of indices of primary fields in (26) which transform under $S_Q$ as the irreducible representation $\lambda$. For example, from (21a), we write

$$\mathcal{S}^{[1] \times [1]} = \mathcal{S}^{[]+[1]+[11]+[2]} = \mathcal{S}^{[]} \cup \mathcal{S}^{[1]} \cup \mathcal{S}^{[11]} \cup \mathcal{S}^{[2]} . \tag{36a}$$

To write down the above spectra, we first define

$$\mathcal{A} = \{ (r,s) \in (\mathbb{N}+5) \times [-1,1) | rs \in \mathbb{Z} \} . \tag{37}$$

Using (16) and (11), we have

$$\mathcal{S}^{[]} = \mathcal{A} \cup \{ \langle 1,1 \rangle^D, \langle 1,2 \rangle^D, (4,0), (4,1) \} , \tag{38a}$$

$$\mathcal{S}^{[1]} = \mathcal{A} \cup \{ (0,1/2), (4,0), (4,\pm 1/2), (4,1) \} , \tag{38b}$$

$$\mathcal{S}^{[11]} = \mathcal{A} \cup \{ (2,\pm 1/2), (4,0), (4,\pm 1/4), (4,\pm 1/2), (4,1) \} , \tag{38c}$$

$$\mathcal{S}^{[2]} = \mathcal{A} \cup \{ (2,0), (2,1), (4,0), (4,\pm 1/4), (4,\pm 1/2), (4,1) \} . \tag{38d}$$

Furthermore, we have used subsets rather than equalities in (34) because some of structure constants in these spectra could vanish non-trivially due to the crossing-symmetry equation. This phenomenon will be demonstrated in some examples of Sections 4 and 5.

## 3.3 Numbers of solutions

The crossing-symmetry equation (24) has a non-trivial number of linearly-independent solutions. For instance, there are four configurations for the four-point connectivities in (4), equivalent to four linearly-independent solutions. Let us then introduce numbers of solutions to the crossing-symmetry equation (24) for the four-point function $\langle \prod_{i=1}^{4} V_{(r_i,s_i)} \rangle$:

$$\mathcal{N}_{\langle \prod_{i=1}^{4} V_{(r_i,s_i)} \rangle} = \dim\{ \text{solutions to (24)} \} , \tag{39}$$

$$\mathcal{N}^{Potts}_{\langle \prod_{i=1}^{4} V_{(r_i,s_i)} \rangle} = \dim\{ \text{solutions to (24) modulo the constraints (34)} \} , \tag{40}$$

which can be counted by using the method of [12]. By its definition, $\mathcal{N}^{\text{Potts}}_{\langle \prod_{i=1}^4 V_{(r_i,s_i)} \rangle}$ is therefore the number of crossing-symmetry solutions that belong to the Potts CFT and always satisfies the inequality:

$$\mathcal{N}^{\text{Potts}}_{\langle \prod_{i=1}^4 V_{(r_i,s_i)} \rangle} \leq \mathcal{N}_{\langle \prod_{i=1}^4 V_{(r_i,s_i)} \rangle}. \tag{41}$$

Several examples will be given in Sections 4.3 and 5 to show that the inequality (41) does not always saturate. On the other hand, the number of solutions $\mathcal{N}^{\text{Potts}}$ can also be deduced by $S_Q$ symmetry. Let us then write

$$\left\langle \prod_{i=1}^4 V^{\lambda_i} \right\rangle = \sum_i T_i F_i, \tag{42}$$

where $T_i$ are $S_Q$ invariant tensors and $F_i$ are solutions to the crossing-symmetry equation (24). The dimension of the linear space spanned by $T_i$, denoted by $\mathcal{I}$, then predicts the number of linearly-independent solutions to (24) and can be computed by using the tensor product (18). From [12], we have

$$\mathcal{I}_{\langle \prod_{i=1}^4 V^{\lambda_i} \rangle} = \sum_\nu M_{\lambda_1,\lambda_2,\nu} M_{\lambda_3,\lambda_4,\nu}, \tag{43}$$

where $M_{\lambda,\mu,\nu}$ are multiplicities in the tensor product (18). From (34), the number $\mathcal{I}_{\langle \prod_{i=1}^4 V^{\Lambda(r_i,s_i)} \rangle}$ then provides an upper bound for (40),

$$\mathcal{N}^{\text{Potts}}_{\langle \prod_{i=1}^4 V_{(r_i,s_i)} \rangle} \leq \mathcal{I}_{\langle \prod_{i=1}^4 V^{\Lambda(r_i,s_i)} \rangle}. \tag{44}$$

## 4 Examples

We discuss solutions to the crossing-symmetry equation (24) for some four-point functions of the Potts CFT in details. Let us start with rewriting the spectrum $\mathcal{S}^{\text{Potts}}$ in (26) with respect to their conformal spins:

$$\mathcal{S}^{\text{odd}} = \{(r,s) \in \mathcal{S}^{\text{Potts}} | rs \in 2\mathbb{Z}+1\}, \tag{45a}$$

$$\mathcal{S}^{\text{even}} = \{(r,s) \in \mathcal{S}^{\text{Potts}} | rs \in 2\mathbb{Z}\} \cup \mathcal{S}^{\text{deg}}, \tag{45b}$$

where $\mathcal{S}^{\text{deg}} = \{\langle 1,1 \rangle^D, \langle 1,2 \rangle^D\}$.

### 4.1 $\langle V_{(0,\frac{1}{2})} V_{(0,\frac{1}{2})} V_{(0,\frac{1}{2})} V_{(0,\frac{1}{2})} \rangle$: The four-point connectivities

The field $V_{(0,\frac{1}{2})}$ belongs to the fundamental representation of $S_Q$. Using (21a), we have the following $s$-channel decomposition:

$$\left\langle V^{[1]}_{(0,\frac{1}{2})} V^{[1]}_{(0,\frac{1}{2})} V^{[1]}_{(0,\frac{1}{2})} V^{[1]}_{(0,\frac{1}{2})} \right\rangle = T_{[]} F^{(s)}_{[]} + T_{[1]} F^{(s)}_{[1]} + T_{[2]} F^{(s)}_{[2]} + T_{[11]} F^{(s)}_{[11]}. \tag{46}$$

To solve for this four-point function, we assume that the input to the crossing-symmetry equation (24) is the spectrum $\mathcal{S}^{\text{Potts}}$ in (6). We then find four linearly independent solutions, which agree with the four-point connectivities in (4) and the decomposition (46). In this case, the crossing-symmetry equation automatically excludes any field that does not transform in irreducible representations of $S_Q$ in (46). To write down the spectra for solutions in (46), since there are 4 solutions in (46), we single out the solution $F^{(s)}_\lambda$ by excluding 3 linearly-independent structure constants of any field which does not transform in the irreducible representation $\lambda$ [12]. For instance, we separate the solution $F^{(s)}_{[2]}$ from the others by requiring vanishing structure constants:

$$D^{(s)}_{\langle 1,1 \rangle^D} = 0, \quad D^{(s)}_{(0,\frac{1}{2})} = 0, \quad \text{and} \quad D^{(s)}_{(2,\frac{1}{2})} = 0. \tag{47}$$

Applying these three constraints on (46) and normalizing one structure constant then give us a unique solution to the crossing-symmetry equation (24). Moreover, the permutation symmetry in the product $V_{(0,\frac{1}{2})} \times V_{(0,\frac{1}{2})}$ constrains $F_{[2]}^{(s)}$ to have only fields with even spins. Let us display the numerical results for some structure constants in the $s$-channel of $F_{[2]}^{(s)}$ at $\beta = 0.8 + 0.1i$:

$$\Delta_{\max} = 30 \qquad\qquad \Delta_{\max} = 60$$

| $(r,s)$ | $\Re D_{(r,s)}^{(s)}$ | deviation | $\Re D_{(r,s)}^{(s)}$ | deviation |
|---|---|---|---|---|
| $(2,1)$ | $0.09662757185$ | $1 \times 10^{-10}$ | $0.09662757185\ldots$ | $7.5 \times 10^{-29}$ |
| $\langle 1,2 \rangle$ | $-2 \times 10^{-10}$ | $0.50$ | $1.0 \times 10^{-27}$ | $0.13$ |
| $(3,0)$ | $1 \times 10^{-12}$ | $0.38$ | $1.0 \times 10^{-28}$ | $0.11$ |
| $(3,\pm\frac{2}{3})$ | $1 \times 10^{-13}$ | $0.90$ | $1.0 \times 10^{-30}$ | $1.0$ |
| $(4,0)$ | $6.9696038 \times 10^{-5}$ | $7.3 \times 10^{-8}$ | $6.9696038\ldots \times 10^{-5}$ | $4.8 \times 10^{-26}$ |
| $(4,\pm\frac{1}{2})$ | $3.5139509 \times 10^{-5}$ | $2.2 \times 10^{-8}$ | $3.5139509\ldots \times 10^{-5}$ | $1.2 \times 10^{-26}$ |

$$(48)$$

where we have chosen the normalization: $D_{(2,0)}^{(s)} = 1$. The structure constants $D_{\langle 1,2 \rangle^D}^{(s)}$, $D_{(3,0)}^{(s)}$, and $D_{(3,\pm\frac{2}{3})}^{(s)}$ in (48) vanish since they do not transform in the representation $[2]$. Moreover, all four-point structure constants in (46) with $r \in 2\mathbb{N}^* + 1$ vanish, which agree with the results of [2]. The spectra for solutions in (46) can be summarized as follows:

$$(49)$$

| Solutions | Spectra | |
|---|---|---|
| | $s$ | $t,u$ |
| $F_{[]}^{(s)}$ | $(\mathcal{S}_{r \in 2\mathbb{N}}^{[]} \cup \mathcal{S}^{\deg}) \cap \mathcal{S}^{\mathrm{even}}$ | |
| $F_{[1]}^{(s)}$ | $\mathcal{S}_{r \in 2\mathbb{N}}^{[1]} \cap \mathcal{S}^{\mathrm{even}}$ | $\mathcal{S}_{r \in 2\mathbb{N}}^{\mathrm{Potts}} \cup \mathcal{S}^{\deg}$ |
| $F_{[11]}^{(s)}$ | $\mathcal{S}_{r \in 2\mathbb{N}}^{[11]} \cap \mathcal{S}^{\mathrm{odd}}$ | |
| $F_{[2]}^{(s)}$ | $\mathcal{S}_{r \in 2\mathbb{N}}^{[2]} \cap \mathcal{S}^{\mathrm{even}}$ | |

The difference between the crossing-symmetry solutions (46) and the four-point connectivities (4) is only a matter of changing bases. They are related by the linear relations:

$$F_{[]}^{(s)} = P_{aaaa} + P_{aabb} + \frac{1}{Q-1}(P_{abab} + P_{abba}), \qquad (50a)$$

$$F_{[1]}^{(s)} = P_{aaaa} + \frac{1}{Q-2}(P_{abab} + P_{abba}), \qquad (50b)$$

$$F_{[2]}^{(s)} = \frac{1}{2}(P_{abab} + P_{abba}), \qquad (50c)$$

$$F_{[11]}^{(s)} = \frac{1}{2}(P_{abab} - P_{abba}), \qquad (50d)$$

where we have normalized the four-point connectivities such that

$$D_{\langle 1,1 \rangle^D}^{aabb} = 1 \quad \text{and} \quad D_{(0,\frac{1}{2})}^{aabb} = -D_{(0,\frac{1}{2})}^{aaaa}. \qquad (51)$$

The linear relations in (50) can be easily computed by comparing the spectra for solutions in (46) with the spectra for the four-point connectivities in [2] and using the analytic ratios in [26]:

$$\frac{D_{(0,\frac{1}{2})}^{aaaa}}{D_{(0,\frac{1}{2})}^{abab}} = -1, \quad \frac{D_{(2,0)}^{aaaa}}{D_{(2,0)}^{abab}} = \frac{2}{2-Q} \quad \text{and} \quad \frac{D_{(2,0)}^{aaaa}}{D_{(2,0)}^{aabb}} = 1-Q. \qquad (52)$$

### 4.1.1 The selection rules

From the numerical results, we conjecture vanishing three-point functions:

$$\langle V_{(0,\frac{1}{2})} V_{(0,\frac{1}{2})} V_{(r,s)}^{\lambda,a} \rangle = 0, \quad \text{for} \quad r \in 2\mathbb{N}^* + 1. \tag{53}$$

For the case $r = 3$, the above trivially follows from (33) since, from (22c) and (22d), primary fields with $r = 3$ only transform in $S_Q$ irreducible representations with three boxes, which do not appear in $[1] \times [1]$. Furthermore, the selection rules (53) do not immediately follow from the spectra in [2] since four-point structure constants can be a sum of the product of three-point structure constants due to non-trivial multiplicities in (16). For instance, the field $V_{(5,0)}^{[2]}$ has multiplicity 2 from (22h). Thus, assuming that the two-point functions of $V_{(5,0)}^{[2],1}$ and $V_{(5,0)}^{[2],2}$ have the same normalization, we write

$$D_{(5,0)}^{[2]} \sim \left( C_{V_{(0,\frac{1}{2})} V_{(0,\frac{1}{2})} V_{(5,0)}^{[2],1}} \right)^2 + \left( C_{V_{(0,\frac{1}{2})} V_{(0,\frac{1}{2})} V_{(5,0)}^{[2],2}} \right)^2. \tag{54}$$

Because these structure constants are complex numbers due to $Q \in \mathbb{C}$, having the four-point structure constants $D_{(5,0)}^{[2]}$ being zero does not imply that three-point structure constants in (54) vanish independently.

In Section 4.3, we nevertheless have checked in 22 examples for the four-point functions $\langle V_{(0,\frac{1}{2})} V_{(0,\frac{1}{2})} V_1 V_2 \rangle$ that the conjecture (53) always holds for crossing-symmetry solutions of the Potts CFT. From (53), we write the fusion rule:

$$V_{(0,\frac{1}{2})} \times V_{(0,\frac{1}{2})} = \sum_{k \in (\mathcal{S}_{r \in 2\mathbb{N}}^{[]} \cup \mathcal{S}^{\text{deg}}) \cap \mathcal{S}^{\text{even}}} V_k^{[]} + \sum_{k \in \mathcal{S}_{r \in 2\mathbb{N}}^{[1]} \cap \mathcal{S}^{\text{even}}} V_k^{[1]} + \sum_{k \in \mathcal{S}_{r \in 2\mathbb{N}}^{[2]} \cap \mathcal{S}^{\text{even}}} V_k^{[2]} + \sum_{k \in \mathcal{S}_{r \in 2\mathbb{N}}^{[11]} \cap \mathcal{S}^{\text{odd}}} V_k^{[11]}.$$

## 4.2 $\langle V_{(0,\frac{1}{2})} V_{(0,\frac{1}{2})} V_{(2,0)} V_{(2,0)} \rangle$ and $\langle V_{(0,\frac{1}{2})} V_{(0,\frac{1}{2})} V_{(2,\frac{1}{2})} V_{(2,\frac{1}{2})} \rangle$

Using (21b) and (21c), $S_Q$ symmetry predicts 5 solutions for both cases, which agree with our findings from the conformal bootstrap. In this case, all crossing-symmetry solutions belong to the Potts CFT. They can be summarized as

| Four-point functions | $s$-channel solutions | $t$-channel solutions |
|---|---|---|
| $\langle V_{(0,\frac{1}{2})} V_{(0,\frac{1}{2})} V_{(2,0)} V_{(2,0)} \rangle$ | $F_{[1]}^{(s)}, F_{[]}^{(s)}, F_{[11]}^{(s)}, F_{[2],0}^{(s)}, F_{[2],1}^{(s)}$ | $F_{[1]}^{(t)}, F_{[2]}^{(t)}, F_{[11]}^{(t)}, F_{[21]}^{(t)}, F_{[3]}^{(t)}$ |
| $\langle V_{(0,\frac{1}{2})} V_{(0,\frac{1}{2})} V_{(2,\frac{1}{2})} V_{(2,\frac{1}{2})} \rangle$ | $G_{[1]}^{(s)}, G_{[]}^{(s)}, G_{[11]}^{(s)}, G_{[2],0}^{(s)}, G_{[2],1}^{(s)}$ | $G_{[1]}^{(t)}, G_{[2]}^{(t)}, G_{[11]}^{(t)}, G_{[21]}^{(t)}, G_{[111]}^{(t)}$ |

(55)

where the $s$-channel and $t$-channel solutions are just different bases for the same space of solutions. To single out each solution in (55), we again impose constraints on their structure constants. For instance, since there are 5 solutions for both four-point functions, we can extract the solutions $F_{[1]}^{(t)}$ and $G_{[1]}^{(t)}$ from (55) by setting 4 structure constants of any primary field that does not transform in the fundamental representation to zero. For example,

$$D_{(2,0)}^{(t)} = D_{(2,\frac{1}{2})}^{(t)} = D_{(3,\frac{1}{3})}^{(t)} = D_{(3,0)}^{(t)} = 0. \tag{56}$$

Therefore, the spectra for each solution in (55) read

| Solutions | Spectra | | |
|---|---|---|---|
| | $t$ | $u$ | $s$ |
| $F_{[3]}^{(t)}$ | $\mathcal{S}^{[3]}$ | | |
| $G_{[111]}^{(t)}$ | $\mathcal{S}^{[111]}$ | | |
| $F_{[21]}^{(t)}, G_{[21]}^{(t)}$ | $\mathcal{S}^{[21]}$ | $\mathcal{S}^{\text{Potts}} - \mathcal{S}^{\text{deg}}$ | $\mathcal{S}_{r \in 2\mathbb{N}}^{\text{Potts}} \cup \{\langle 1,1 \rangle^D\}$ |
| $F_{[2]}^{(t)}, G_{[2]}^{(t)}$ | $\mathcal{S}^{[2]}$ | | |
| $F_{[11]}^{(t)}, G_{[11]}^{(t)}$ | $\mathcal{S}^{[11]}$ | | |
| $F_{[1]}^{(t)}, G_{[1]}^{(t)}$ | $\mathcal{S}^{[1]}$ | | |
| $F_{[]}^{(s)}, G_{[]}^{(s)}$ | | | $(\mathcal{S}_{r \in 2\mathbb{N}}^{[]} \cup \{\langle 1,1 \rangle^D\}) \cap \mathcal{S}^{\text{even}}$ |
| $F_{[1]}^{(s)}, G_{[1]}^{(s)}$ | | | $\mathcal{S}_{r \in 2\mathbb{N}}^{[1]} \cap \mathcal{S}^{\text{even}}$ |
| $F_{[11]}^{(s)}, G_{[11]}^{(s)}$ | $\mathcal{S}^{\text{Potts}} - \mathcal{S}^{\text{deg}}$ | $\mathcal{S}^{\text{Potts}} - \mathcal{S}^{\text{deg}}$ | $\mathcal{S}_{r \in 2\mathbb{N}}^{[11]} \cap \mathcal{S}^{\text{odd}}$ |
| $F_{[2],0}^{(s)}, G_{[2],0}^{(s)}$ | | | $\mathcal{S}_{r \in 2\mathbb{N}}^{[2]} \cap \mathcal{S}^{\text{even}} - \{(2,0)\}$ |
| $F_{[2],1}^{(s)}, G_{[2],1}^{(s)}$ | | | $\mathcal{S}_{r \in 2\mathbb{N}}^{[2]} \cap \mathcal{S}^{\text{even}} - \{(2,1)\}$ |

$$(57)$$

From the fusion rules (25), only the degenerate fields with $r \in 2\mathbb{N}^* + 1$ are allowed in the $s$-channel while all degenerate fields are subtracted from the $t, u$-channels. Moreover, for each four-point function, solutions that transform in the representation $[2]$ form a two-dimensional subspace of solutions whose bases can be chosen arbitrarily. For example, we write down each of their bases by excluding one of the fields $V_{(2,0)}$ and $V_{(2,1)}$. Let us now deduce the fusion rules of $V_{(0,\frac{1}{2})} \times V_{(2,0)}$ and $V_{(0,\frac{1}{2})} \times V_{(2,\frac{1}{2})}$,

$$V_{(0,\frac{1}{2})} \times V_{(2,0)} = \sum_{k \in \mathcal{S}^{[3]}} V_k^{[3]} + \sum_{k \in \mathcal{S}^{[21]}} V_k^{[21]} + \sum_{k \in \mathcal{S}^{[2]}} V_k^{[2]} + \sum_{k \in \mathcal{S}^{[11]}} V_k^{[11]} + \sum_{k \in \mathcal{S}^{[1]}} V_k^{[1]}, \qquad (58)$$

and

$$V_{(0,\frac{1}{2})} \times V_{(2,\frac{1}{2})} = \sum_{k \in \mathcal{S}^{[111]}} V_k^{[111]} + \sum_{k \in \mathcal{S}^{[21]}} V_k^{[21]} + \sum_{k \in \mathcal{S}^{[2]}} V_k^{[2]} + \sum_{k \in \mathcal{S}^{[11]}} V_k^{[11]} + \sum_{k \in \mathcal{S}^{[1]}} V_k^{[1]}. \qquad (59)$$

We have checked for several examples in Section (4.3) that the above fusion rules always agree with crossing-symmetry solutions for the four-point functions $\langle V_{(0,\frac{1}{2})} V_{(2,0)} V_1 V_2 \rangle$ and $\langle V_{(0,\frac{1}{2})} V_{(2,\frac{1}{2})} V_1 V_2 \rangle$ of the Potts CFT.

### 4.3 More examples

We first define $L = \sum_{i=1}^4 r_i$ for $\langle \prod_{i=1}^4 V_{(r_i,s_i)} \rangle$. Let us then count the numbers of crossing-symmetry solutions, defined in (39) and (40), and also compute the prediction from $S_Q$ symmetry in (43) for 28 four-point functions with $L \leq 6$.

For convenience, these four-point functions are labelled as their indices. In 17 out of 28 cases, we find solutions that do not belong to the Potts CFT. Moreover, $\mathcal{N}^{\text{Potts}}$ aways obeys the inequality (44) and saturates the bound from $S_Q$ symmetry in 24 out of 28 cases.

$0 \leq L \leq 2$,

| Four-point functions | $\mathcal{N}$ | $\mathcal{N}^{\text{Potts}}$ | $\mathcal{I}$ |
|:---:|:---:|:---:|:---:|
| $(0, \frac{1}{2})^4$ | 4 | 4 | 4 |
| $(0, \frac{1}{2})^3 (2, 0)$ | 3 | 3 | 3 |
| $(0, \frac{1}{2})^3 (2, \frac{1}{2})$ | 3 | 3 | 3 |
| $(0, \frac{1}{2})^3 (2, 1)$ | 3 | 3 | 3 |

$L = 3$,

| Four-point functions | $\mathcal{N}$ | $\mathcal{N}^{\text{Potts}}$ | $\mathcal{I}$ |
|:---:|:---:|:---:|:---:|
| $\left(0, \frac{1}{2}\right)^3 (3, 0)$ | 5 | 2 | 2 |
| $\left(0, \frac{1}{2}\right)^3 (3, \frac{1}{3})$ | 5 | 2 | 2 |

$L = 4$,

| Four-point functions | $\mathcal{N}$ | $\mathcal{N}^{\text{Potts}}$ | $\mathcal{I}$ |
|:---:|:---:|:---:|:---:|
| $(0, \frac{1}{2})^3 (4, 0)$ | 12 | 6 | 14 |
| $(0, \frac{1}{2})^3 (4, \frac{1}{2})$ | 12 | 6 | 13 |
| $(0, \frac{1}{2})^3 (4, \frac{1}{4})$ | 12 | 6 | 6 |
| $(0, \frac{1}{2})^2 (2, 0)^2$ | 5 | 5 | 5 |
| $(0, \frac{1}{2})^2 (2, 1)^2$ | 5 | 5 | 5 |
| $(0, \frac{1}{2})^2 (2, \frac{1}{2})^2$ | 5 | 5 | 5 |
| $(0, \frac{1}{2})^2 (2, \frac{1}{2})(2, -\frac{1}{2})$ | 5 | 5 | 5 |
| $(0, \frac{1}{2})^2 (2, 0)(2, \frac{1}{2})$ | 4 | 4 | 4 |
| $(0, \frac{1}{2})^2 (2, 1)(2, \frac{1}{2})$ | 4 | 4 | 4 |
| $(0, \frac{1}{2})^2 (2, 0)(2, 1)$ | 5 | 5 | 5 |

$L = 5$,

| Four-point functions | $\mathcal{N}$ | $\mathcal{N}^{\text{Potts}}$ | $\mathcal{I}$ |
|:---:|:---:|:---:|:---:|
| $\left(0, \frac{1}{2}\right)^2 (2, 0)(3, 0)$ | 9 | 5 | 5 |
| $\left(0, \frac{1}{2}\right)^2 (2, \frac{1}{2})(3, 0)$ | 9 | 5 | 5 |
| $\left(0, \frac{1}{2}\right)^2 (2, 0)(3, \frac{1}{3})$ | 9 | 5 | 5 |
| $\left(0, \frac{1}{2}\right)^2 (2, \frac{1}{2})(3, \frac{1}{3})$ | 9 | 5 | 5 |

$L = 6$,

| Four-point functions | $\mathcal{N}$ | $\mathcal{N}^{\text{Potts}}$ | $\mathcal{I}$ |
|---|---|---|---|
| $(0, \frac{1}{2})(2, 0)^3$ | 8 | 7 | 7 |
| $(0, \frac{1}{2})(2, \frac{1}{2})^3$ | 8 | 7 | 7 |
| $(0, \frac{1}{2})(2, \frac{1}{2})^2(2, 0)$ | 8 | 7 | 7 |
| $(0, \frac{1}{2})(2, \frac{1}{2})^2(2, -\frac{1}{2})$ | 8 | 7 | 7 |
| $(0, \frac{1}{2})(2, 0)^2(2, \frac{1}{2})$ | 8 | 7 | 7 |
| $(0, \frac{1}{2})(2, -\frac{1}{2})(2, \frac{1}{2})(2, 0)$ | 8 | 7 | 7 |
| $(0, \frac{1}{2})^2(3, 0)^2$ | 15 | 8 | 12 |
| $(0, \frac{1}{2})^2(3, \frac{1}{3})^2$ | 15 | 8 | 11 |

**Numbers of solutions and the degenerate fields**

From our results for the $O(n)$ CFT in [12], observe from several examples that the numbers of solutions do not seem to depend on the second Kac indices of four-point functions, whenever their spectra do not have any degenerate field. For instance, the numbers of solutions for $\langle V_{(\frac{1}{2},0)} V_{(\frac{1}{2},0)} V_{(\frac{3}{2},\frac{2}{3})} V_{(\frac{3}{2},0)} \rangle^{O(n)}$ and $\langle V_{(\frac{1}{2},0)} V_{(\frac{1}{2},0)} V_{(\frac{3}{2},\frac{2}{3})} V_{(\frac{3}{2},-\frac{2}{3})} \rangle^{O(n)}$, whose spectra contain no degenerate fields, coincide but differ from $\langle V_{(\frac{1}{2},0)} V_{(\frac{1}{2},0)} V_{(\frac{3}{2},\frac{2}{3})} V_{(\frac{3}{2},\frac{2}{3})} \rangle^{O(n)}$, which has the identity field $V_{\langle 1,1 \rangle}^D$ in the $s$-channel.

In the Potts CFT, the same observations still hold. However, even if there are degenerate fields in some channels, the numbers of solutions for four-point functions with the same $r_i$ but different $s_i$ may still coincide due to the degenerate field $V_{\langle 1,2 \rangle}^D$, which does not exist in the $O(n)$ CFT. For example, the numbers of solutions for $\langle V_{(0,\frac{1}{2})} V_{(0,\frac{1}{2})} V_{(2,\frac{1}{2})} V_{(2,\frac{1}{2})} \rangle$ and $\langle V_{(0,\frac{1}{2})} V_{(0,\frac{1}{2})} V_{(2,\frac{1}{2})} V_{(2,-\frac{1}{2})} \rangle$ match because the fusion rules (25) allow the degenerate field $V_{\langle 1,2 \rangle}^D$ to propagate in the $s$-channel of $\langle V_{(0,\frac{1}{2})} V_{(0,\frac{1}{2})} V_{(2,\frac{1}{2})} V_{(2,-\frac{1}{2})} \rangle$. Let us now propose the following conjecture:

**Conjecture for both the Potts and $O(n)$ CFTs:** *If there are no degenerate fields in all three channels, the numbers of crossing-symmetry solutions for the four-point functions $\langle \prod_{i=1}^4 V_{(r_i,s_i)} \rangle$ are independent of $s_i$.*

For the Potts CFT, the conjecture is for both $\mathcal{N}$ and $\mathcal{N}^{\text{Potts}}$. For example, observe that the four-point functions $\langle V_{(0,\frac{1}{2})} \prod_{i=1}^3 V_{(2,j_i)} \rangle$ come with $\mathcal{N} = 8$ and $\mathcal{N}^{\text{Potts}} = 7$, regardlessly of $j_i$.

## 5 Examples with extra solutions

We discuss some examples in Section 4.3 with $\mathcal{N}^{\text{Potts}} < \mathcal{N}$ and show how to pin down which solutions belong to the Potts CFT.

### 5.1 $\langle V_{(0,\frac{1}{2})} V_{(0,\frac{1}{2})} V_{(0,\frac{1}{2})} V_{(3,0)} \rangle$ and $\langle V_{(0,\frac{1}{2})} V_{(0,\frac{1}{2})} V_{(0,\frac{1}{2})} V_{(3,\frac{1}{3})} \rangle$

These are the simplest cases where we have extra solutions. In both cases, we find 5 linearly-independent solutions to the crossing-symmetry equation (24), whereas $S_Q$ representation theory predicts only two solutions.

For the case $V_{(3,0)}$, there are in fact two different fields: $V_{(3,0)}^{[3]}$ and $V_{(3,0)}^{[111]}$ from (22c). We then write

$$[3] \times [1] = [4] + [31] + [3] + [21] + [2], \tag{60a}$$
$$[111] \times [1] = [1111] + [211] + [111] + [11]. \tag{60b}$$

Therefore, using (60) with (21a), the four-point functions $\langle V_{(0,\frac{1}{2})} V_{(0,\frac{1}{2})} V_{(0,\frac{1}{2})} V_{(3,0)}^{[3]} \rangle$ and $\langle V_{(0,\frac{1}{2})} V_{(0,\frac{1}{2})} V_{(0,\frac{1}{2})} V_{(3,0)}^{[111]} \rangle$ transform under $S_Q$ symmetry as $[2] + [11]$ in all three channels. These two four-point functions can then be built from solutions of which spectra contains only fields that can be decomposed into $[2]$ or $[11]$. There are only two of these solutions, which can be obtained by requiring vanishing structure constants:

$$D_{(3,0)}^{(s)} = D_{(3,0)}^{(t)} = D_{(3,0)}^{(u)} = 0. \tag{61}$$

In other words, there are three other solutions, in which structure constants in (61) do not vanish. These three solutions will be discussed in Section 6. The two solutions that belong to the Potts CFT have the following spectra:

$$\left\langle V_{(0,\frac{1}{2})} V_{(0,\frac{1}{2})} V_{(0,\frac{1}{2})} V_{(3,0)}^{\lambda} \right\rangle$$

| $\lambda$ | Spectra for $s, t, u$ |
|---|---|
| $[111]$ | $\mathcal{S}_{r \in 2\mathbb{N}}^{[11]} \cap \mathcal{S}^{\text{odd}}$ |
| $[3]$ | $\mathcal{S}_{r \in 2\mathbb{N}}^{[2]} \cap \mathcal{S}^{\text{even}}$ |

$$\tag{62}$$

The field $V_{(3,\frac{1}{3})}$ transforms under $S_Q$ as the irreducible representation $[21]$. Therefore, using (20) and (21a), the four-point function $\langle V_{(0,\frac{1}{2})} V_{(0,\frac{1}{2})} V_{(0,\frac{1}{2})} V_{(3,\frac{1}{3})}^{[21]} \rangle$ can be decomposed into $[2] + [11]$ in all three channels. Similarly, we find that there are 2 out of 5 solutions which fit with such decomposition. They again satisfy (61) and come with the spectra:

$$\left\langle V_{(0,\frac{1}{2})} V_{(0,\frac{1}{2})} V_{(0,\frac{1}{2})} V_{(3,\frac{1}{3})}^{[21]} \right\rangle$$

| Solutions | Spectra | |
|---|---|---|
| | $s$ | $t, u$ |
| $F_{[11]}^{(s)}$ | $\mathcal{S}_{r \in 2\mathbb{N}}^{[11]} \cap \mathcal{S}^{\text{odd}}$ | $\mathcal{S}_{r \in 2\mathbb{N}}^{[11]} \cup \mathcal{S}_{r \in 2\mathbb{N}}^{[2]}$ |
| $F_{[2]}^{(s)}$ | $\mathcal{S}_{r \in 2\mathbb{N}}^{[2]} \cap \mathcal{S}^{\text{even}}$ | |

$$\tag{63}$$

## 5.2 $\langle V_{(0,\frac{1}{2})} V_{(2,0)} V_{(2,0)} V_{(2,0)} \rangle$

Using (21d) and (21b), all three channels of this four-point function can be decomposed into

$$[3] + 2[21] + 2[2] + [11] + [1]. \tag{64}$$

That is to say $S_Q$ symmetry predicts 7 linearly-independent solutions. However, the bootstrap approach finds 8 solutions. It turns out that only 7 out of 8 solutions have spectra which fit

with the decomposition (64). The spectra of these 7 solutions are given by

| Solutions | Spectra | |
|---|---|---|
| | $s$ | $t, u$ |
| $F_{[3]}^{(s)}$ | $\mathcal{S}^{[3]} \cap \mathcal{S}^{\text{even}}$ | $\mathcal{S}^{\text{Potts}} - \mathcal{S}^{\text{deg}} - \{(3,1)\}$ |
| $F_{[21],\text{even}}^{(s)}$ | $\mathcal{S}^{[21]} \cap \mathcal{S}^{\text{even}}$ | |
| $F_{[2],0}^{(s)}$ | $\mathcal{S}^{[2]} \cap \mathcal{S}^{\text{even}} - \{(2,0)\}$ | |
| $F_{[2],1}^{(s)}$ | $\mathcal{S}^{[2]} \cap \mathcal{S}^{\text{even}} - \{(2,1)\}$ | |
| $F_{[1]}^{(s)}$ | $\mathcal{S}^{[1]} \cap \mathcal{S}^{\text{even}}$ | |
| $F_{[11]}^{(s)}$ | $\mathcal{S}^{[11]} \cap \mathcal{S}^{\text{odd}}$ | $\mathcal{S}^{\text{Potts}} - \mathcal{S}^{\text{deg}} - \{(3,1),(3,\pm 1/3)\}$ |
| $F_{[21],\text{odd}}^{(s)}$ | $\mathcal{S}^{[21]} \cap \mathcal{S}^{\text{odd}}$ | $\mathcal{S}^{\text{Potts}} - \mathcal{S}^{\text{deg}} - \{(3,1),(2,\pm 1/2)\}$ |

$$(65)$$

where the degenerate fields are subtracted due to the fusion rules (25). In other words, 7 solutions in (65) can be exacted from requiring

$$D_{(3,1)}^{(s)} = D_{(3,1)}^{(t)} = D_{(3,1)}^{(u)} = 0 \,. \tag{66}$$

Vanishing structure constants in (66) do not contradict with the fusion rule (58) but are consequences of the permutation symmetry of the OPE, $V_{(2,0)} \times V_{(2,0)}$, which allows each spectrum in (65) to have either odd or even spins [12]. In this case, the crossing-symmetry equation prefers the spectrum $\mathcal{S}^{[3]} \cap \mathcal{S}^{\text{even}}$, over the spectrum $\mathcal{S}^{[3]} \cap \mathcal{S}^{\text{odd}}$. Moreover, the eighth solution comes with the spectra:

$$\mathcal{S}^{(s)} = \mathcal{S}^{\text{Potts}} \cap \mathcal{S}^{\text{odd}} - \{(2,1/2),(3,1/3)\} \,, \tag{67a}$$

$$\mathcal{S}^{(t,u)} = \mathcal{S}^{\text{Potts}} - \mathcal{S}^{\text{deg}} \,. \tag{67b}$$

Let us also stress here that while the fields $V_{(2,\frac{1}{2})}$ and $V_{(3,\frac{1}{3})}$ are excluded in (67a), the fields $V_{(2,-\frac{1}{2})}$ and $V_{(3,-\frac{1}{3})}$ indeed appear in (67a). That is to say, structure constants in the solution with the spectra (67a) and (67b) do not obey the relation, $D_{(r,-s)} = D_{(r,s)}$, in contrast to the other 7 solutions in (65) where such relation always holds.

Since solutions in different channels are different choices of bases for the same space of solutions to the crossing-symmetry equation (24), one can then check that all solutions in (65) belong to the same space of solutions by numerically computing the linear relations of solutions in the $s$- and $t$- channel on the table (65),

$$F_\lambda^{(t)} = \sum_\mu \alpha_\lambda{}^\mu F_\mu^{(s)} \,. \tag{68}$$

We find that $\alpha_\lambda{}^\mu$ exist for any solution in (65) and do not vanish, except for $\alpha_{[11]}^{[21],\text{odd}}$. Notice that having $\alpha_{[11]}^{[21],\text{odd}}$ being zero is consistent with the $t$- and $u$- channel spectra of the solutions $F_{[11]}^{(s)}$ and $F_{[21],\text{odd}}^{(s)}$. This ensures us that the eighth solution lives in a different space of solution, and all of 7 solutions in (65) are indeed in the same space of solutions of the Potts CFT.

### 5.3 $\langle V_{(0,\frac{1}{2})} V_{(0,\frac{1}{2})} V_{(0,\frac{1}{2})} V_{(4,0)} \rangle$

In this case, $S_Q$ symmetry predicts 14 solutions, and we find 12 solutions to the crossing-symmetry equation. However, only 6 out of these 12 solutions belong to the Potts CFT.

For this example, there are 9 four-point functions from (22e). However, since the tensor products $[1] \times [4]$, $[1] \times [211]$, and $[1] \times [22]$ can only be decomposed into representations with more than three boxes, none of which appear in $[1] \times [1]$, we have

$$\langle V_{(0,\frac{1}{2})} V_{(0,\frac{1}{2})} V_{(0,\frac{1}{2})} V_{(4,0)}^{[4]} \rangle = \langle V_{(0,\frac{1}{2})} V_{(0,\frac{1}{2})} V_{(0,\frac{1}{2})} V_{(4,0)}^{[211]} \rangle = \langle V_{(0,\frac{1}{2})} V_{(0,\frac{1}{2})} V_{(0,\frac{1}{2})} V_{(4,0)}^{[22]} \rangle = 0 \,. \tag{69}$$

The other 6 four-point functions can be decomposed into $S_Q$ representations as follows:

$$\left\langle V_{(0,\frac{1}{2})} V_{(0,\frac{1}{2})} V_{(0,\frac{1}{2})} V_{(4,0)}^{\lambda} \right\rangle$$

| $\lambda$ | Mul | $S_Q$ representations in $s,t,u$ |
|---|---|---|
| $[3]$ | 1 | $[2]$ |
| $[21]$ | 1 | $[2]+[11]$ |
| $[2]$ | 2 | $[2]+[11]+[1]$ |
| $[1]$ | 1 | $[2]+[11]+[1]+[\,]$ |
| $[\,]$ | 1 | $[1]$ |

(70)

Only 6 out of 12 solutions fit with the decompositions in (70). We extract these 6 solutions by imposing 6 constraints on 12 solutions,

$$D_{(3,0)}^{(s)} = D_{(3,0)}^{(t)} = D_{(3,0)}^{(u)} = 0 \quad \text{and} \quad D_{(3,\frac{1}{3})}^{(s)} = D_{(3,\frac{1}{3})}^{(t)} = D_{(3,\frac{1}{3})}^{(u)} = 0 \,. \tag{71}$$

The total spectra for these 6 solutions are $\mathcal{S}_{r \in 2\mathbb{N}}^{\text{Potts}}$. To single out each solution, we introduce

$$\mathcal{S}_{(r,s)} = \mathcal{S}_{r \in 2\mathbb{N}+6}^{\text{Potts}} \cup \{(r,j) \mid j = \pm s, \ \pm(s-1) \text{ and } j \in (-1,1]\} \,. \tag{72}$$

The above is equivalent to removing 5 linearly-independent structure constants with $r \le 4$ in the spectrum $\mathcal{S}_{r \in 2\mathbb{N}}^{\text{Potts}}$. We then introduce the crossing-symmetry solutions $F_{(r,s)}^{(s)}$ which have $\mathcal{S}_{(r,s)}$ for the s-channel spectrum and $\mathcal{S}_{r \in 2\mathbb{N}}^{\text{Potts}}$ for the $t$- and $u$-channel spectra. As an example, let us also display the deviation of some structure constants for $F_{(4,\frac{1}{4})}^{(s)}$ at $\beta = 0.8 + 0.1i$:

<table>
<tr><td></td><td></td><td colspan="2">$\Delta_{\max} = 20$</td><td>$\Delta_{\max} = 40$</td></tr>
<tr><td>$(r,s)$</td><td>ch</td><td colspan="2">deviation</td><td>deviation</td></tr>
<tr><td>$(0,\frac{1}{2})$</td><td>$t$</td><td colspan="2">$4.1 \times 10^{-6}$</td><td>$2.6 \times 10^{-14}$</td></tr>
<tr><td>$(2,0)$</td><td>$t$</td><td colspan="2">$5.9 \times 10^{-6}$</td><td>$6.3 \times 10^{-14}$</td></tr>
<tr><td>$(2,1)$</td><td>$t$</td><td colspan="2">$5.9 \times 10^{-6}$</td><td>$6.1 \times 10^{-14}$</td></tr>
<tr><td>$(4,\pm\frac{1}{4})$</td><td>$s$</td><td colspan="2">$4.4 \times 10^{-6}$</td><td>$2.3 \times 10^{-14}$</td></tr>
<tr><td>$(4,\pm\frac{3}{4})$</td><td>$s$</td><td colspan="2">$5.2 \times 10^{-7}$</td><td>$7.3 \times 10^{-14}$</td></tr>
</table>

(73)

The 6 linearly-independent solutions in (71) are

$$F_{(0,\frac{1}{2})}^{(s)} \,, F_{(2,0)}^{(s)} \,, \quad F_{(2,\frac{1}{2})}^{(s)} \,, \quad F_{(4,0)}^{(s)} \,, \quad F_{(4,\frac{1}{2})}^{(s)} \quad \text{and} \quad F_{(4,\frac{1}{4})}^{(s)} \,. \tag{74}$$

Four-point functions in (70) can then be written as linear combinations of solutions in (74) by

imposing constraints on some of their structure constants:

$$\left\langle V_{(0,\frac{1}{2})} V_{(0,\frac{1}{2})} V_{(0,\frac{1}{2})} V_{(4,0)}^{\lambda} \right\rangle$$

| $\lambda$ | Mul | Vanishing $D_{(r,s)}$ |
|-----------|-----|------------------------|
| [3] | 1 | $D_{(0,\frac{1}{2})}^{(s)}, D_{(2,\frac{1}{2})}^{(s)}, D_{(4,\frac{1}{4})}^{(s)}, D_{(0,\frac{1}{2})}^{(t,u)}, D_{(2,\frac{1}{2})}^{(t,u)}$ |
| [21] | 1 | $D_{(0,\frac{1}{2})}^{(s)}, D_{(0,\frac{1}{2})}^{(t,u)}$ |
| [2] | 2 | — |
| [1] | 1 | — |
| [] | 1 | $D_{(2,0)}^{(s)}, D_{(2,\frac{1}{2})}^{(s)}, D_{(4,\frac{1}{4})}^{(s)}, D_{(2,0)}^{(t,u)}, D_{(2,\frac{1}{2})}^{(t,u)}$ |

(75)

The four-point functions $\langle V_{(0,\frac{1}{2})} V_{(0,\frac{1}{2})} V_{(0,\frac{1}{2})} V_{(4,0)}^{[3]} \rangle$ and $\langle V_{(0,\frac{1}{2})} V_{(0,\frac{1}{2})} V_{(0,\frac{1}{2})} V_{(4,0)}^{[]} \rangle$ can then be completely fixed up to a normalization factor because they are linear combinations of 6 solutions in (74) in which we require 5 vanishing structure constants, while the linear combination of solutions for the four-point function $\langle V_{(0,\frac{1}{2})} V_{(0,\frac{1}{2})} V_{(0,\frac{1}{2})} V_{(4,0)}^{[21]} \rangle$ has 3 unfixed coefficients since we only need 2 structure constants to vanish. For the cases of $\lambda = [2]$ and $[1]$, the three linear combinations are linearly dependent and cannot be fixed with our current method.

# 6 Conclusion and outlook

We demonstrate how to numerically solve the crossing-symmetry equation for several four-point functions of primary fields in the Potts CFT, from which we conclude some exact results such as their numbers of crossing-symmetry solutions, their spectra, and some of their fusion rules. Our results also support that the Potts CFT is consistent with the spectrum of [8]. For instance, the number of crossing-symmetry solutions $\mathcal{N}^{\text{Potts}}$ is always consistent with the prediction from $S_Q$ symmetry in all of our examples wherein the inequality $\mathcal{N}^{\text{Potts}} \leq \mathcal{I}$ always holds and becomes saturated in 24 out of 28 cases. Similarly to the $O(n)$ CFT [12], the discrepancy between $\mathcal{N}^{\text{Potts}}$ and $\mathcal{I}$ suggests that the Potts CFT may have a larger global symmetry than $S_Q$. Let us now point out possible future directions:

**Crossing-symmetry solutions $\leftrightarrow$ Physical Observables?**

From the lattice model [2], the field $V_{(0,\frac{1}{2})}(z)$ inserts a Fortuin-Kasteleyn cluster at point $z$, as shown in (4) for the four-point connectivities, whereas the other non-diagonal fields $V_{(r,s)}(z)$ with $r \neq 0$ insert $2r$ lines at point $z$, as boundaries for clusters. This gives us a glimpse that more general four-point functions of the Potts CFT should describe clusters with more complicated geometry. However, in general, we do not know yet the relations between crossing-symmetry solutions and these observables of the lattice model.

A first step towards such relations is perhaps to assume that there is always a one-to-one correspondence between solutions to the crossing-symmetry equation and configurations of clusters, similarly to the case of the four-point connectivities in (50). Then try to look for rules of drawing clusters for each four-point function in 4.3 such that the number of their configurations always matches with $\mathcal{N}^{\text{Potts}}$.

**Extra solutions $\rightarrow$ New CFT?**

There is a significant number of extra solutions to the crossing-symmetry equation (24) that do not belong to Potts CFT. Let us now display the simplest examples of these extra solutions.

The four-point function $\langle V_{(0,\frac{1}{2})} V_{(0,\frac{1}{2})} V_{(0,\frac{1}{2})} V_{(3,0)} \rangle$ comes with three extra solutions:

| Solutions | Spectra for $s,t,u$ |
|---|---|
| $\mathcal{X}^1$ | $\mathcal{S}^{\text{ex}} - \{(2,0)\}$ |
| $\mathcal{X}^2$ | $\mathcal{S}^{\text{ex}} - \{(2,1)\}$ |
| $\mathcal{X}^3$ | $\mathcal{S}^{\text{ex}} - \{(4,0)\}$ |

(76)

where $\mathcal{S}^{\text{ex}} = \mathcal{S}^{\text{Potts}} \cap \mathcal{S}^{\text{even}} - \mathcal{S}^{\text{deg}}$. Similarly to (76), by our set-up, the other extra solutions on the tables in Section 4.3 always have spectra which are subsets of the spectrum of the Potts CFT and also have vast intersections with the spectrum of the $O(n)$ CFT in [12]. Nevertheless, these extra solutions belong to neither CFTs since they do not fit with the constraints from $S_Q$ symmetry in (34) and also come with the field $V_{(0,\frac{1}{2})}$ that does not exist in the $O(n)$ CFT. Let us suggest some plausible explanations:

- It could be that some of these extra solutions belong to a bigger CFT whose spectrum contains the spectra of the Potts and $O(n)$ CFTs as subsets. For example, we find solutions to the crossing-symmetry equation for four-point functions which mix primary fields from both the Potts and $O(n)$ CFT, e.g. $\langle V_{(0,\frac{1}{2})} V_{(0,\frac{1}{2})} V_{(1,0)} V_{(1,0)} \rangle$ where the field $V_{(1,0)}$ does not exist in the Potts CFT but appears in the spectrum of the $O(n)$ CFT. It may be interesting to investigate further to see if such big CFT exists.

- Since we are solving the crossing-symmetry equation for four-point structure constants, it is therefore not clear whether four-point structure constants of these extra solutions always factorize into products of three-point structure constants. If not, they do not belong to a consistent CFT. One way of clarifying this issue is to consider the crossing-symmetry equation as a system of quadratic equations for three-point structure constants.

**Analytic structure constants**

Likewise to the results of [26] and [11] for the four-point connectivities, it is also possible to deduce analytic ratios of some structure constants in more general four-point functions from our numerical results. For example, let us display some ratios for structure constants of $V_{(4,0)}^{\lambda}$ and $V_{(4,\frac{1}{2})}^{\lambda}$ with $\lambda = [1],[3]$ in the $t,u$-channel of the four-point function $\langle V_{(2,0)} V_{(2,0)} V_{(0,\frac{1}{2})} V_{(0,\frac{1}{2})} \rangle$:

$$\frac{D_{(4,0)}^{[3]}}{D_{(4,0)}^{[2]}} = \frac{10(Q-1)(Q-2)(Q-6)}{2Q^3 - 15Q^2 + 29Q - 18}, \tag{77a}$$

$$\frac{D_{(4,0)}^{[3]}}{D_{(4,0)}^{[1]}} = \frac{2(Q-2)(Q-6)(Q^2 - 4Q + 2)}{Q(Q-1)(Q-4)^2}, \tag{77b}$$

$$\frac{D_{(4,\frac{1}{2})}^{[3]}}{D_{(4,\frac{1}{2})}^{[2]}} = \frac{2(Q-2)^3}{Q^2}, \tag{77c}$$

$$\frac{D_{(4,\frac{1}{2})}^{[3]}}{D_{(4,\frac{1}{2})}^{[1]}} = \frac{2(Q-2)^3}{(Q-4)^2(Q-1)}, \tag{77d}$$

which hold for generic $Q \in \mathbb{C}$ at high precision. However, we have not yet found any analytic formula for a single structure constant as the three-point connectivity in [27]. Understanding

the general structure of these analytic ratios is certainly a crucial step towards solving the Potts CFT.

**Other observables**

The authors of [28] considered another kind of physical observables in the critical Potts model, the spin-cluster connectivities. The four-point spin connectivities can be described by the four-point function $\langle V_{(\frac{1}{2},0)} V_{(\frac{1}{2},0)} V_{(\frac{1}{2},0)} V_{(\frac{1}{2},0)} \rangle$ whose channels can have a non-degenerate diagonal field with the dimensions $(\Delta_{(1,2)}, \Delta_{(1,2)})$ propagating. Their complete spectra have not yet been found. Finding crossing-symmetry solutions which fit this description should therefore be interesting

# Acknowledgements

I am very grateful to Linnea Grans-Samuelsson, Jesper Lykke Jacobsen, Sylvain Ribault, and Hubert Saleur for many useful discussions and collaborating on related projects, especially S. Ribault who provided many suggestions on writing this article.
Many thanks to Raoul Santachiara for carefully commenting on the draft and stimulating discussions on related subjects. I am also grateful for all three referees of SciPost for several useful comments.

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
