# Peer review of "Global symmetry and conformal bootstrap in the two-dimensional $Q$-state Potts model"

_SciPost Physics, doi:SciPost Phys. 14, 155 (2023)_

## Round 1 · Referee Report · Anonymous (Referee 1) · 2023-2-22

Strengths

New insights on a very difficult and important problem.

Weaknesses

Written for people already working on the subject. On the other hand, I think this choice is the best one: there are other related papers with full of detail that have to be read to understand this. For the people working in this problem, it is convenient to find in a direct and not diluted way the main results .

Report

This paper belongs to a series of papers, on an on-going fruitful research, that will provide text-book results in two-dimensional physics. I recommend for publication.

---

## Round 1 · Referee Report · Anonymous (Referee 2) · 2023-2-22

Report

The paper can be published as it is.

---

## Round 1 · Referee Report · Anonymous (Referee 3) · 2023-3-1

Report

The paper can be published as it is.

---

## Round 1 · List of Changes

I am very grateful to the editor and reviewers for their work. Please find below answers to comments/questions of Reports 2 and 3.

Answers to comments and questions from Report 3:

%1-The linearly independent four-point connectivities were first considered in arXiv:1104.4323. The author could add a reference to this paper.

Answer to %1:

I have referred to this paper at the introduction of the four-point connectivities in (1.4).

%2-Can the author explain how the number of crossing symmetric solutions to the bootstrap equations without the $S_Q$ constraints is obtained? Is it possible to obtain the dimension of this vector space without solving the bootstrap equations?

Answer to %2:

For each four-point function, solving the crossing-symmetry equation with the spectrum of the Potts CFT gives us a linear combination of solutions. Then we look into each of these solutions whether their spectra satisfy the $S_Q$ constraint (3.12). In general, we find that not all of them satisfies (3.12). I have attempted to summarize this concept in "Main results". Examples of how to extract these extra solutions are also presented in Section 5.

We do not know yet how to count the extra solutions without solving the crossing-symmetry equation.

However, in the critical $O(n)$ model, we now understand how to count solutions without solving the crossing-symmetry equation by counting two-dimensional graphs, known as the combinatorial maps. This new approach of counting solutions will appear in our upcoming paper. It should be interesting to find a similar approach which also works with the Potts model. If succeed, it will allow us to count the extra solutions.

%3-Are the structure constants obtained analytic functions of $c$? In principle, I think, in its geometrical representation the model considered makes sense only for $c\leq 1$

Answer to %3

Yes, structure constants are analytic function of the central charge, and we expect that the structure constants themselves are valid for generic value of the central charge in (1.1).

For instance, in arxiv:2007.04190, we have tested that the formula for the structure constants $D_{(0, 1/2)}$ in the four-point connectivities from arxiv:1009.1314 coincide with the numerical bootstrap at very high precision for complex value of the central charge.

This also agrees with the fact that the Fortuin-Kasteleyn description of the Potts model makes sense for generic $Q$.

%4-The additional not $S_Q$ invariant solutions are present also for integer $0\leq Q\leq 4$?

Answer to %4:

For integer $Q$, we expect that the extra solutions would still appear. For instance, results for these special values of $Q$ could be obtained by taking limits of our results for generic $Q$, which are solutions to the linear system (3.2). In general, a limit should not change the number of solutions to linear system. This situation should indeed be studied in more details by considering the limits $Q$ approaching integers of these extra solutions.

%5-A final curiosity: the papers, arXiv:1008.1216 and arXiv:1111.4033 presented a construction of spin clusters in the $Q$-state Potts model at arbitrary $Q$. Are some of the fields that appear there, also present in the spectrum of [2]?

Answer to %5:

The critical exponents in (3) of 1008.1218 is related to conformal dimension (2.1) as follows

$h_{l_1 - l_2, 2*l_1} = \Delta_{2*l_1, l_1 - l_2} $

where $\kappa$ in (4) of 1008.1218 is our $4\beta^2$.

Since $\Delta_{r, s}$ with $r \in 2\mathbb{N}$ and $s \in \mathbb{{Z}$, so the critical exponents of 1008.1218 appear in the spectrum (2.2) for integer $l_1$ and $l_2$. It might be interesting to investigate further if some of our bootstrap solutions are observables of the spin-cluster Potts model in 1008.1218

Answers to Report 2:

%1. The introduction is extremely technical. I would recommend a rewriting to make it clearer what the author would like to show/prove and to contextualize the study. For instance in the last paragraph, it is written: "more general four point functions...", it would be good to make more precise statements and put them into context.

%2. The section on main results, while could be quite interesting due to the technical level of the paper, is not very illuminating since it is mostly a table of content. Also here, "the simplest 28 four point functions" is a bit arbitrary, either the author explains which are the simplest before or it should be specified which class has been considered. I would recommend either to remove the section and put it in the concluding section as a summary or to rewrite it completely to make more clear what are the main results.

Answer to %1 and %2:

Regarding these two points, I have attempted to write a less technical introduction by referring to some physical models on the table (1.2). I have also rewritten "Main results" and added more specific details.

Answer to %3:

%3. In eq (3.1) how can one compute the coefficients D? Later on it is written that they are all computable and it would be nice to give an (even short) account of how to do it.

Coefficients D in the inter-chiral block expansion (3.1) can be considered as products of three-point structure constants, which can be completely determined by using the BPZ equation, the crossing-symmetry equation, and the single-valuedness of four-point functions of the types: $<V_{1,2}^D V_1 V_2 V_3 >$ and $<V_{1,2}^D V_1 V_{1,2}^D V_1 >$ where $V_{1,2}^D$ is the level-2 degenerate field and $V_1, V_2 V_3$ are generic primary fields.

I have added this explanation below (3.1), as well as a reference for examples of detailed calculations of $D$ in (3.1).

%4. After eq. (3.2), two comments:
%4.1-why the intermediate states need to be the same in all channels?
%4.2-the comment regarding having three relations to avoid infinitely many solutions sounds very interesting but it is not clear what it means and how to interpret it. I would like the author to clarify it.

Answer to %4.1:

For generic four-point functions, the intermediate states are not the same in all channels.

In particular, the spectra $S^{(s)}$, $S^{(t)}$, $S^{(u)}$ are the full spectrum of the Potts model allowed by the degenerate fusion rules (3.2). So $S^{(s)}$, $S^{(t)}$, $S^{(u)}$ are identical for four-point functions of identical fields but not for generic four-point functions. This ensures us the resulting solutions are at least all crossing-symmetry solutions in the Potts model, as will be demonstrated for several examples in Sections 4 and 5.

I have added this explanation above (3.3), as well as an example of four-point functions whose the intermediate states are not the same below (3.7).

Answer to %4.2:

We do not know yet the interpretation of having infinitely many solutions, and I have clarified that this issue is still an open problem below (3.2).

%5. In the examples sections it seems that the author is inputting the exact spectrum into the crossing relations to compute the three point function. Am I right? If so, how the procedure is carried over needs to be discussed before eq. (3.8), meaning in that subsection.

Answer to %5

Not quite, we compute the four-point structure constants $D$ by inputting the exact spectrum into the crossing-symmetry equation.

However, we could deduce three-point functions from our method. For instance, we wrote down the fusion rules (4.14) and (4.15) by checking that (4.14) and (4.15) always hold for many examples of four-point functions of the type $<V_{2,1/2}V_{0, 1/2}V_1V_2$ and $<V_{2,0}V_{0, 1/2}V_1V_2>$ of the Potts model.

This clarification has been added below (4.14) and (4.15).

---

## Editorial Decision

published